# Glioblastoma Multiforme Tumors in Women Have a Lower Expression of Fatty Acid Elongases *ELOVL2*, *ELOVL5*, *ELOVL6*, and *ELOVL7* than in Men

**DOI:** 10.3390/brainsci12101356

**Published:** 2022-10-06

**Authors:** Jan Korbecki, Donata Simińska, Dariusz Jeżewski, Klaudyna Kojder, Patrycja Tomasiak, Maciej Tarnowski, Dariusz Chlubek, Irena Baranowska-Bosiacka

**Affiliations:** 1Department of Biochemistry and Medical Chemistry, Pomeranian Medical University in Szczecin, Powstańców Wielkopolskich 72, 70-111 Szczecin, Poland; 2Department of Neurosurgery and Pediatric Neurosurgery, Pomeranian Medical University in Szczecin, Unii Lubelskiej 1, 71-252 Szczecin, Poland; 3Department of Applied Neurocognitivistics, Pomeranian Medical University in Szczecin, Unii Lubelskiej 1, 71-252 Szczecin, Poland; 4Department of Anaesthesiology and Intensive Care, Pomeranian Medical University in Szczecin, Unii Lubelskiej 1, 71-281 Szczecin, Poland; 5Institute of Physical Culture Sciences, University of Szczecin, Piastów 40b/6, 71-065 Szczecin, Poland; 6Department of Physiology in Health Sciences, Pomeranian Medical University in Szczecin, Żołnierska 54, 70-210 Szczecin, Poland

**Keywords:** cancer, fatty acid, elongase, MUFA, PUFA, brain tumor

## Abstract

One line of research on the possible ways of inhibiting the growth of glioblastoma multiforme (GBM), a brain tumor with a very poor prognosis, is the analysis of its metabolism, such as fatty acid synthesis by desaturases and elongases. This study examines the expression of elongases *ELOVL1*, *ELOVL2*, *ELOVL3*, *ELOVL4*, *ELOVL5*, *ELOVL6*, and *ELOVL7* in GBM tumor samples from 28 patients (16 men and 12 women), using a quantitative real-time polymerase chain reaction (qRT-PCR). To demonstrate the influence of the tumor microenvironment on the tested elongases, U-87 MG cells were cultured in nutrient-deficient conditions and with cobalt chloride (CoCl_2_) as a hypoxia-mimetic agent. The results showed that the expression of *ELOVL1* and *ELOVL7* in the GBM tumor was lower than in the peritumoral area. The expression of six of the seven studied elongases differed between the sexes. Hypoxia increased the expression of *ELOVL5* and *ELOVL6* and decreased the expression of *ELOVL1*, *ELOVL3*, *ELOVL4*, and *ELOVL7* in U-87 MG cells. These results indicate that the synthesis of fatty acids, especially polyunsaturated fatty acids (PUFA), in GBM tumors may be higher in men than in women. In contrast, the synthesis of saturated fatty acids (SFA) may be higher in women than in men.

## 1. Introduction

Glioblastoma multiforme (GBM) is a grade IV glioma [1], with an estimated incidence of 3–5 cases per 100,000 people; it is 1.2 to 2.6 times more common in men than in women [2]. Current therapy for this cancer consists of surgical intervention, with radiotherapy in combination with chemotherapy [3,4]. A current therapeutic approach to GBM treatment based on the use of a DNA alkylating agent (temozolomide) is inadequate and highly ineffective, as shown by the very poor prognosis for newly diagnosed patients, giving an average survival time of only 10 months [5], with only about 5% of patients surviving more than 5 years after diagnosis [6].

One direction of research on finding an effective treatment for GBM is to gain a thorough understanding of metabolism, such as fatty acid synthesis in GBM tumors. In human cells, de novo fatty acid synthesis occurs with saturated fatty acids (SFA) and monounsaturated fatty acids (MUFA) [7]. The first stage mainly involves the synthesis of palmitic acid C16:0 by fatty acid synthase (FAS) and acetyl-CoA carboxylase (ACC) [7,8]. Subsequently, palmitic acid C16:0 becomes the substrate for the production of other SFAs and MUFAs. In contrast, the synthesis of major polyunsaturated fatty acids (PUFAs) in humans does not occur de novo. The precursors for n-3 and n-6 PUFAs are linolenic acid (ALA) 18:3n-3 and linoleic acid (LA) 18:2n-6, respectively. The conversion of fatty acids to other fatty acids depends on desaturases and elongases [7,9]. Desaturases form a double bond and elongases lengthen the chain by two carbons. The elongation of a hydrocarbon chain by two carbons in a fatty acid follows four reactions: condensation, reduction, dehydration, and reduction. The first of these, condensation, is catalyzed by elongases and represents the rate-controlling step of fatty acid elongation by two carbons [7,9]. Mammals have 7 elongases—elongation of long-chain fatty acid family members 1–7—(Elovl)1–7. Elovl6 is responsible for the synthesis of stearic acid C18:0 from palmitic acid C16:0, and the second synthesis reaction of cis-vaccenic acid C18:1n-7 from palmitoleic acid C16:1n-7 [10,11,12]. Elovl1, Elovl3, and Elovl7 are responsible for further elongation of the hydrocarbon chain into stearic acid C18:0 [7,9,13,14]. Elovl2 and Elovl5 are responsible for PUFA biosynthesis [15,16]. Elovl2, but not Elovl5, is responsible for the biosynthesis of DHA C22:6n-3 and other very long-chain PUFAs in the last steps of this fatty acid synthesis. Elovl5 is responsible for the elongation of MUFAs, but exhibits little activity against C16:1n-7 compared to Elovl6 [15]. Elovl4 is responsible for the synthesis of very long-chain fatty acids with lengths greater than 26 carbons [17,18].

In a previous study, we analyzed changes in the expression of fatty acid desaturases in GBM and found a reduction in the expression of *SCD* and *FADS2* in GBM tumors, with no differences in the expression of fatty acid desaturases between women and men [19]. In order to have a thorough understanding of the fatty acid synthesis pathways in GBM, it is necessary to study the expression of elongases, especially in relation to the gender of the GBM tumor patients—men have a worse prognosis and lower survival rate than women [20,21]. Therefore, the aim of this study was to analyze the expression of elongases involved in fatty acid biosynthesis.

## 2. Materials and Methods

### 2.1. Patient Samples

The tumor samples used in this study were obtained in 2014 by the Department of Biochemistry and the Department of Neurosurgery and Pediatric Neurosurgery of the Pomeranian Medical University in Szczecin in a project that addressed purinergic receptors in GBM progression. This project was accepted by the local bioethical commission (KB-0012/96/14), and the study was conducted in accordance with the Declaration of Helsinki.

Tumor samples were collected from 28 adult patients (16 men and 12 women) with GBM tumors during surgery at the Department of Neurosurgery and Pediatric Neurosurgery of the Pomeranian Medical University in Szczecin, Poland (Table 1 and Figure 1). Patients were initially diagnosed with brain tumors using magnetic resonance imaging (MRI) or computed tomography (CT). Each patient had then been recommended for neurosurgery following the radiological diagnosis. During the brain tumor removal, surgery patients underwent standard general anesthesia with endotracheal intubation. During the neuronavigation procedure, craniotomy and tumor resections were performed according to the classical method (bone removal and dura incision, tumor visualization, resection, biopsy for histopathological and molecular examination, closure of the dura, bone restoration in some patients, subcutaneous tissue and skin closure in some patients). During surgery, the results of the 1.5 Tesla (T) and 3T MRI were fed into the computer station of a neuronavigation device to estimate the position of the surgical instruments in relation to the GBM tumor with the help of a video camera recording the surgical movement (precision of 2–3 mm). During surgery, a biopsy was performed for a histopathological examination to confirm the brain tumor was grade IV, as defined by the WHO [1].

Clinical radiological morphology made it possible to distinguish three tumor zones commonly used in the literature and clinical practice (Figure 2) [22,23]:The peritumoral area, a buffer zone between the GBM tumor and healthy tissue, with individual foci of infiltration. The peritumoral area is considered an appropriate control for the study of GBM tumors [24]. For this reason, the results obtained from GBM tumor studies were assimilated into the peritumoral area.The non-enhancing tumor core. This area of GBM is usually located in the central part of the GBM tumor,The enhancing tumor region, surrounding the tumor core.

### 2.2. Cell Cultures and Treatment

U-87 MG cells from the European Collection of Authenticated Cell Cultures (ECACC) were used for in vitro experiments. These cells were cultured in Eagle’s minimum essential medium (EMEM) (Sigma-Aldrich, Poznań, Poland) supplemented with 10% (*v*/*v*) heat-inactivated fetal bovine serum (FBS) (Gibco Limited, Poznań, Poland), 2 mM L-glutamine, 1 mM sodium pyruvate (Sigma-Aldrich, Poznań, Poland), 1% non-essential amino acids (Sigma-Aldrich, Poznań, Poland), and antibiotics (100 U/mL penicillin, and 100 µg/mL streptomycin (Gibco Limited, Poznań, Poland)) at 37 °C in a humidified atmosphere of 95% air and 5% CO_2_. Cells were lavaged twice weekly with 0.25% trypsin-ethylenediaminetetraacetic acid (EDTA) solution (Sigma-Aldrich, Poznań, Poland).

An in vitro study was performed to investigate the effect of the tumor microenvironment on the elongases studied. In some areas of the GBM tumor are structures called pseudopalisades [25]. These structures are significantly distant from blood vessels, or the blood vessels in such a structure are obstructed, leading to hypoxia and nutrient deficiency. To emulate this, U-87 MG cells would be cultured under hypoxic or nutrient-deficient conditions. First, test cells were seeded onto 6-well plates at a density of 20,000 cells/cm^2^. After 3 days of incubation, the medium was changed and the U-87 MG cells were divided and cultured under two different conditions for 24 h: control (standard culture medium), or hypoxic or nutrient-deficient. The control cells were incubated on a standard culture medium. To mimic hypoxic conditions, 200 mM of cobalt chloride (CoCl_2_) (Sigma-Aldrich, Poznań, Poland) was used. This is a commonly used hypoxia-mimetic agent [26]. To produce nutrient deficiency, U-87 MG cells were incubated with a low concentration of L-glutamine (0.2 mM) and without sodium pyruvate (volume supplemented with PBS), leaving 1.0 g/L (5.5 mM) of glucose in the medium. These conditions have been described by us and applied in a previous work in which we studied the expression of fatty acid desaturases in GBM [19]. After 24 h of incubation under those conditions, the U-87 MG cells were trypsinized, and the resulting cell pellet was washed with PBS solution and centrifuged again (25 °C, 300 G, 5 min). The cell pellet was used for RNA analysis.

### 2.3. Quantitative Real-Time Polymerase Chain Reaction (qRT-PCR)

Analysis of the elongase expression from patient material and U-87 MG cells was performed using two-step reverse transcription PCR (RT-PCR). First, RNA extraction was performed from the patient material using RNeasy Lipid Tissue Mini Kit (Qiagen, Hilden, Germany), and from the cellular material using RNeasy Mini Kit (Qiagen). Then, cDNA was produced using a First Strand cDNA synthesis kit and oligo-dT primers (Fermentas, Waltham, MA, USA). Analysis of selected mRNAs was performed using an ABI 7500 Fast instrument with Power SYBR Green PCR Master Mix reagent (Applied Biosystems, Waltham, MA, USA). Real-time conditions were as follows: 95 °C for 15 s, 40 cycles at 95 °C for 15 s, and 60 °C for 1 min. C_t_ values were used for further analysis. The quantity of the test transcript was normalized to an endogenous control, namely, *glyceraldehyde-3-phosphate dehydrogenase* (*GAPDH*) gene, was calculated as the fold difference (2^dCt), and further processed using statistical analysis. Data were presented as tumor tissue absolute expression. *GAPDH* is considered the reference gene for GBM studies [27]. The following primer pairs were used:for *GAPDH*, (5′-TCATGGGTGTGAACCATGAGAA-3′ and 5′-GGCATGGACTGTGGTCATGAG-3′);for *ELOVL1*, (5′-TTATTCTCCGAAAGAAAGACGGG-3′ and 5′-ATGACATGCACGGAAGAGTTTAT-3′)for *ELOVL2*, (5′-ATGTTTGGACCGCGAGATTCT-3′ and 5′-CCCAGCCATATTGAGAGCAGATA-3′);for *ELOVL3*, (5′-CTGTTCCAGCCCTATAACTTCG-3′ and 5′-GAATGAGGTTGCCCAATACTCC-3′);for *ELOVL4*, (5′-GAGCCGGGTAGTGTCCTAAAC-3′ and 5′-CACACGCTTATCTGCGATGG-3′);for *ELOVL5*, (5′-TAACAGGAGTATGGGAAGGCA-3′ and 5′-ACCAGAGGACACGGATAATCTT-3′);for *ELOVL6*, (5′-AACGAGCAAAGTTTGAACTGAGG-3′ and 5′-TCGAAGAGCACCGAATATACTGA-3′);for *ELOVL7*, (5′-GCCTTCAGTGATCTTACATCGAG-3′ and 5′-AGGACATGAGGAGCCAATCTT-3′).

### 2.4. Statistical Methods

The expression levels of the elongases tested in this paper are reported in relation to the reference gene, GAPDH, as an average of all the results obtained for a given group. All statistical analyses were performed using Statistica software (version 13, StatSoft Polska, Kraków, Poland). Values of *p* ≤ 0.05 will be considered statistically significant. A Shapiro–Wilk test was performed and showed that the results did not have a normal distribution, so non-parametric tests were used for statistical analyses: Wilcoxon signed-rank tests for comparisons between tumor zones and between culture conditions of the U-87 MG cells; Mann–Whitney U-tests for comparisons between groups of patients; Spearman rank correlation coefficients for the analysis of correlations between the gene expression in each of the three zones, and between gene expressions and the data given by the patients in the questionnaire.

The study with 28 patients (16 men and 12 women) had sufficient statistical power to detect with 80% probability the following real effect sizes for the analyzed quantitative variables: difference between tumor zones or culture conditions (paired measurements) corresponding to 0.55 standard deviations of the difference between paired measurements for the analyzed parameter; difference between males and females (independent groups) corresponding to 1.2 standard deviations of the analyzed parameter; correlation between analyzed parameters corresponding to the value of correlation coefficient equal to ±0.50.

## 3. Results

### 3.1. ELOVL1 and ELOVL7 Expression Was Lower in the GBM Tumor Than in the Peritumoral Area

Analysis of elongase expression showed that *ELOVL1* and *ELOVL7* expressions were lower in the GBM tumor than in the peritumoral area (Figure 3). In the tumor core, the expression of *ELOVL1* (*p* = 0.04) and *ELOVL7* (*p* = 0.013) was lower than in the peritumoral area. The same results were obtained when analyzing the enhancing tumor region where *ELOVL1* (*p* = 0.0013) and *ELOVL7* (*p* = 0.004) expressions were lower than in the peritumoral area.

### 3.2. Expression of Elongases in GBM Tumors Differed between Genders

The *ELOVL1* expression in the tumor core relative to the peritumoral area was greater in the women (*p* = 0.0016) and lower in the men (*p* = 0.0016) (Figure 4).

The *ELOVL2* expression was significantly higher in the peritumoral area in women compared to the men (*p* = 0.013). In contrast, the *ELOVL2* expression in the women was lower in GBM tumors (enhancing tumor region—*p* = 0.02; tumor core—*p* = 0.004) than in the peritumoral area. In the enhancing tumor region, the expression of *ELOVL2* in the women was lower than in the peritumoral area (*p* = 0.04). In the men, the expression level of *ELOVL2* was the same in all the examined regions of the GBM tumor.

The *ELOVL3* expression was not different in the GBM tumor core relative to the peritumoral area in both the women and men (*p* > 0.05).

The *ELOVL4* expression was elevated in the GBM tumor in the women. In the enhancing tumor region relative to the peritumoral area, the *ELOVL4* expression was significantly higher (*p* = 0.02).

The *ELOVL5* expression in the peritumoral area did not differ between genders. In the tumor, there was decreased expression of *ELOVL5* in the women compared to the peritumoral area (enhancing tumor region—*p* = 0.015; tumor core—*p* = 0.005), and an increased expression of *ELOVL5* in the men (enhancing tumor region—*p* = 0.03; tumor core—*p* = 0.055). In the enhancing tumor region and tumor core, the *ELOVL5* expression in the women was lower than in the men (enhancing tumor region—*p* = 0.02; tumor core—*p* = 0.04).

The *ELOVL6* expression in the peritumoral area did not differ between the genders. The expression of *ELOVL6* in the women was lower (*p* = 0.03) in the tumor core than in the peritumoral area, and was lower than in the men (*p* = 0.04).

The *ELOVL7* expression was lower in the GBM tumor in women. In the enhancing tumor region (*p* = 0.004) and tumor core (*p* = 0.00007), the expression of *ELOVL7* was lower than in the peritumoral area.

### 3.3. Expression of Elongases in the GBM Tumors Correlated with Each Other

To a large extent, the expression of elongases positively correlated with each other in the enhancing tumor region and tumor core (Table 2). The expression of *ELOVL7* was with *ELOVL6*, *ELOVL7* with *ELOVL5*, *ELOVL7* with *ELOVL4*, *ELOVL6* with *ELOVL5*, *ELOVL6* with *ELOVL4*, *ELOVL6* with *ELOVL2*, *ELOVL5* with *ELOVL4*, *ELOVL5* with *ELOVL4*, *ELOVL5* with *ELOVL2*, and *ELOVL4* with *ELOVL2*. The *ELOVL1* expression was weakly correlated with the other elongases, as was the *ELOVL3* expression, which positively correlated with the other elongases. The expression of individual elongases between the different GBM tumor areas studied was weakly correlated. The *ELOVL4* expression in the peritumoral area was positively correlated with the tumor core. Additionally, the expression of *ELOVL7* in the peritumoral area was negatively correlated with the expression of this elongase in the enhancing tumor region. In contrast, the positive correlation of the expression of the elongases in question between the enhancing tumor region with the tumor core sometimes occurs. The *ELOVL2* expression in the tumor core was positively correlated with the *ELOVL1* expression from the enhancing tumor region. Additionally, the *ELOVL3* expression in the tumor core was positively correlated with the *ELOVL5* expression from the enhancing tumor region and negatively correlated with the *ELOVL2* expression from the enhancing tumor region. Additionally, the *ELOVL5* expression in the tumor core was positively correlated with the *ELOVL6* expression in the enhancing tumor region.

There was a positive correlation between *ELOVL4*, *ELOVL5*, *ELOVL6*, and *ELOVL7* expressions in the GBM tumor (Table 3). The ELOVL3 expression in the same region of the GBM was not correlated with other elongases except for the ELOVL2 expression in the enhancing tumor region. Additionally, the *ELOVL1* expression in the tumor core was positively correlated with *ELOVL4*, *ELOVL5*, *ELOVL6*, and *ELOVL7* expressions. Within the GBM tumor, there was a positive correlation of expression of only *ELOVL2* in the tumor core with the enhancing tumor region. There was a positive correlation in elongase expression between the enhancing tumor region and the tumor core in only two cases. There was a negative correlation in the *ELOVL2* expression in the enhancing tumor region with the *ELOVL3* expression in the tumor core. Additionally, there was a positive correlation in the *ELOVL2* expression in the tumor core with the *ELOVL6* expression in the enhancing tumor region.

In the GBM tumors in the women, there was a positive correlation in the expression of *ELOVL7* with *ELOVL5*, *ELOVL7* with *ELOVL4*, *ELOVL6* with *ELOVL5*, *ELOVL6* with *ELOVL3*, *ELOVL6* with *ELOVL2*, *ELOVL5* with *ELOVL4*, and *ELOVL5* with *ELOVL2* (Table 4). There was also a positive correlation of *ELOVL2* with *ELOVL3*, *ELOVL4*, *ELOVL5*, *ELOVL6*, and *ELOVL7*. The expression of the same elongase was rarely correlated between the tumor core and the enhancing tumor regions. The *ELOVL3* expression in the tumor core was negatively correlated with *ELOVL5* and *ELOVL6* expressions in the enhancing tumor region. 

### 3.4. The Expression of Elongases in the GBM Tumors Correlated with BMI and Cigarette Pack-Years in the Men, and Correlated with Age and Body Weight in the Women

In the tumor core, the expression of *ELOVL3* had a negative correlation with body weight and BMI, while in the peritumoral area, there was a negative correlation between the *ELOVL2* expression and height and a negative correlation between the *ELOVL4* expression and cigarette pack-years (Table 5). In contrast, in the enhancing tumor region, the *ELOVL2* expression positively correlated with the number of cigarette packs smoked per year, the *ELOVL4* expression positively correlated with height, the *ELOVL7* expression negatively correlated with age, and the *ELOVL5* expression positively correlated with body weight and BMI.

In the men, the expression of the elongases positively correlated with BMI (Table 4)—in particular, the *ELOVL2* expression in the peritumoral area and tumor core, and *ELOVL5* and *ELOVL6* expressions in the enhancing tumor region. There was a negative correlation between the *ELOVL3* expression in the tumor core and BMI. Additionally, in the men, there was a positive correlation between the *ELOVL2* expression in the enhancing tumor region and the number of cigarette packs smoked per year, and a negative correlation between *ELOVL1*, *ELOVL4* and *ELOVL7* expressions in the tumor core and the number of cigarette packs smoked per year. There was a positive correlation of the *ELOVL7* expression in the peritumoral area with body weight.

In the women, there was a negative correlation between *ELOVL5* and *ELOVL6* expressions in the GBM tumor and BMI. Additionally, the expression of *ELOVL2* and *ELOVL5* in the enhancing tumor region negatively correlated with age in the women. The expression of *ELOVL7* in the enhancing tumor region was negatively correlated with cigarette packs smoked per year. In the tumor core, the expression of *ELOVL1* was positively correlated with cigarette packs smoked per year and negatively correlated with age, body weight, and BMI.

### 3.5. Expression of ELOVL5 and ELOVL6 Increased under the Influence of Hypoxia, and the Expression of ELOVL1, ELOVL3, ELOVL4, and ELOVL7 Decreased

The hypoxia-mimetic agent CoCl_2_ caused an increase in *ELOVL5* and *ELOVL6* expressions in the U-87 MG cells by approximately 35% relative to the controls (Figure 5). This indicates that hypoxia increases the expression of *ELOVL5* (*p* = 0.00003) and *ELOVL6* (*p* = 0.00003) in GBM cells. Additionally, in the same model, hypoxia reduced the expression of *ELOVL1* (*p* = 0.0026), *ELOVL3* (*p* = 0.02), *ELOVL4* (*p* = 0.00005), and *ELOVL7* (*p* = 0.0002) by approximately 50%. The expression of *ELOVL2* was not analyzed because it was below the detection limit in U-87 MG cells.

Nutrient-deficient conditions reduced the expression of *ELOVL1* (*p* = 0.0018), *ELOVL4* (*p* = 0.0018), and *ELOVL7* (*p* = 0.010) in the U-87 MG cells. Nutrient-deficient conditions in a slight but statistically significant way decreased the expression of *ELOVL5* (*p* = 0.04). Nutrient-deficient conditions did not significantly affect the expression of *ELOVL3* or *ELOVL6* (*p* > 0.05).

## 4. Discussion

### 4.1. Expression of Elongases in GBM Tumors

To date, only a few studies have analyzed the expression of elongases in GBM tumors, and their results vary widely. In our study, we demonstrated that the expression of *ELOVL2*, *ELOVL3*, *ELOVL4*, *ELOVL5*, and *ELOVL6* was similar in the GBM tumor and in the peritumoral area, confirming the results of Seifert et al. (2015) [28]. Saurty-Seerunghen et al. (2019) showed that *ELOVL2* undergoes increased expression in GBM tumors [29]. Vyazunova et al. (2014) [30] showed an increased expression of ELOVL6 in murine GBM.

We showed that *ELOVL1* and *ELOVL7* expressions were lower in the GBM tumor than in the peritumoral area. This is partly consistent with the results of Seifert et al. (2015) who showed that the *ELOVL7* expression is lower in GBM tumors [28]. Additionally, Kaplan et al. showed that there is less behenic acid C22:0 in GBM tumors than in grade II gliomas [31]. This confirms our results that the synthesis of very-long-chain SFA in GBM tumors was reduced as a result of a decreased expression of *ELOVL1* and *ELOVL7*. At the same time, *ELOVL3* was also responsible for the synthesis of very-long-chain SFA. Nevertheless, our analysis showed that the expression of *ELOVL3* was about 100 times lower than *ELOVL1* and 10 times lower than *ELOVL7*. In some samples, *ELOVL3* expression was below the detection threshold, which indicates that *ELOVL1* and *ELOVL7* were responsible for the synthesis of very-long-chain SFA in GBM tumors.

### 4.2. Gender Differences in the Expression of Elongases

Our study is the first research on gender-related differences in the expression of elongases in GBM. We showed that the expression of *ELOVL2*, *ELOVL5*, and *ELOVL6* in the GBM tumor was lower in women than in men. In contrast, the expression of *ELOVL1* and *ELOVL4* was upregulated, and *ELOVL7* was downregulated in GBM tumors in women. Our results indicate that in GBM tumors, men have higher PUFA synthesis and lower very-long-chain SFA synthesis than women. Additionally, in GBM tumors in men, the synthesis of stearic acid and oleic acid was higher than in women. The *ELOVL2* expression in the peritumoral area was higher in the women than in the men. In the peritumoral area, the expression of *ELOVL5* and *ELOVL6* was the same for each gender. As this is the first demonstration of such correlations in a GBM tumor, it is very difficult to relate our findings to the literature data.

The gender difference in the expression of the studied elongases may indicate the possible influence of sex hormones. The regulatory mechanism of elongase genes has to date been partially understood in the liver, a key organ involved in the biosynthesis of PUFA, and are then transported to other organs, including the brain [32,33]. In hepatocytes, Elovl6 activity has been shown to be increased by estrogens [34,35] and decreased by androgens [34,36]. The *ELOVL2* and *ELOVL5* expressions are not altered by estrogens [33,37,38].

The estrogen-mediated regulation of MUFA and PUFA synthesis takes place by altering the expression of desaturases *SCD*, *FADS1*, and *FADS2* [33,37,38,39]. As PUFA are only produced in the liver [32,33], the sex hormones can only act on the expression of enzymes involved in fatty acid synthesis in the liver, and not in the brain [33]. This may explain the lack of difference in the *ELOVL5* expression in the brain between men and women [40,41].

The *ELOVL2* expression was higher in the peritumoral area in women compared to men. *ELOVL2* is a gene that has two estrogen response elements (EREs) in its promoter [42], a fact which may explain the increased expression of *ELOVL2* in women in the peritumoral area compared to men. However, the average age of the female subjects at the time of surgery in our study was 60.8 years. Therefore, most of the women studied may have had reduced estrogen levels compared to premenopausal women [43] and the effect of estrogen on the *ELOVL2* expression in the brain of the women included in our study may have been low. At the same time, the expression of *ELOVL2* and *ELOVL5* in the enhancing tumor region in women negatively correlated with their age. That indicates that the age of the women influences the expression of elongases involved in PUFA biosynthesis in GBM.

The observed differences may depend on the expression of the epithelial growth factor receptor (EGFR). Men have a higher frequency of EGFR mutations in GBM tumors than women [44]. These mutations are also associated with the amplification of this gene. In contrast, studies on genes associated with inherited GBM risk indicate that EGFR is associated with GBM incidence in men, but not in women [45]. EGFR encodes a receptor for EGF, which is significant for GBM tumor development [46]. It activates sterol regulatory element-binding protein 1 (SREBP-1) through the PI3K-Akt/PKB pathway [47]. As SREBP-1 causes an increased expression of *ELOVL5* [48,49], *ELOVL6* [50,51], and *ELOVL7* [52], men may have a higher expression of *ELOVL5*, *ELOVL6*, and *ELOVL7* in GBM tumors. This may explain our results where women had a lower expression of *ELOVL5*, *ELOVL6*, and *ELOVL7* in GBM tumors compared to men.

Regarding gender differences in *ELOVL1* and *ELOVL4* expressions, there are no literature data on the effect of sex hormones on the expression of these elongases. For this reason, we cannot refer to the literature data.

Differences in elongase expressions between men and women may also partly depend on patient BMI. The expression of elongases in the liver, particularly *ELOVL5* and *ELOVL6*, is higher under the influence of a fatty diet and obesity, as shown by experiments on mice [51]. This may explain the correlation between the *ELOVL5* expression in the enhancing tumor region and BMI in all patients. The same correlation was also found in men, i.e., BMI vs. *ELOVL5* and *ELOVL6* expressions in the enhancing tumor region and the *ELOVL2* expression in the tumor core. This effect may be gender-dependent. In the female GBM tumors, the *ELOVL5* and *ELOVL6* expressions negatively correlated with BMI. This shows that in women, obesity and overweight are associated with a decreased expression of *ELOVL5* and *ELOVL6* in GBM tumors, whereas in men, the expression of these elongases is increased.

### 4.3. Effects of Hypoxia and Nutrient-Deficient Conditions on the Expression of Elongases

In the present study, hypoxia induced by CoCl_2_ increased the expression of *ELOVL5* and *ELOVL6* but decreased the expression of *ELOVL1*, *ELOVL3*, *ELOVL4*, and *ELOVL7* in the U-87 MG cells. These results are consistent with the literature data. Indeed, hypoxia has been shown to increase SREBP-1 activity in vitro [53]. Those cited studies were also performed on the U-87 MG line, as in this study. Additionally, HIF-1-dependent hypoxia increases SREBP expression as shown by a study on breast cancer cells [54]. SREBP-1 is a transcription factor that increases the expression of genes involved in lipid synthesis, particularly fatty acids [55], as well as *FAS*, *ACC*, and *SCD* [55]. SREBP also elevates the expression of *ELOVL6* [50,51], which confirms our results in which CoCl_2_ increased the *ELOVL6* expression. The same mechanism (hypoxia SREBP-1) may be responsible for the regulation of the hypoxia-induced increase in the *ELOVL5* expression. This is associated with the fact that the expression of *ELOVL5* is also dependent on SREBP-1c [48,49]. To date, there has been no literature data on the effect of hypoxia on the expression of *ELOVL1*, *ELOVL3*, *ELOVL4*, and *ELOVL7*, so we cannot refer our data to other studies.

Our results are the first to show the effect of the hypoxia mimetic agent on the expression of these elongases. The elongase expression results show that hypoxia decreases the synthesis of very-long-chain SFA but increases the synthesis of MUFA and PUFA. This indicates that under hypoxia, the cell membrane begins to be remodeled into a more fluid membrane (less very-long-chain SFA, more PUFA with more double bonds) which may have a role in signal transduction from various membrane receptors.

In our study, nutrient deficiency decreased *ELOVL1*, *ELOVL4*, *ELOVL5*, and *ELOVL7* expressions in U-87 MG cells, but not that of *ELOVL3* and *ELOVL6*. This effect was much smaller than that observed under hypoxic conditions. It is difficult to relate our results to the literature data because the expression of elongases such as *ELOVL5* is poorly understood. So far, it has been shown that the expression of *ELOVL5* is rarely different. The *ELOVL5* expression, compared to *ELOVL6*, is not altered by SREBP-1c, glucose, or diabetes [51]. In contrast, the *ELOVL5* expression is increased by PPARα activation and a high-fat diet [51]. PPARα is a transcription factor involved in the expression of genes related to fatty acid oxidation [56], that is, fatty acid catabolism. The nutrient deficiency used in this study is associated with insufficient substances to produce the optimal amount of fatty acids and a reduction in fatty acid oxidation. This may explain why in our results the nutrient deficiency condition decreased the *ELOVL5* expression in U-87 MG cells. We have shown that nutrient deficiency reduces the *ELOVL4* expression in U-87 MG cells. The regulation of the *ELOVL4* expression has been poorly understood. It is known that diabetes reduces the *ELOVL4* expression in the retina [57]. Diabetes is a disease caused by decreased insulin action which reduces the uptake and metabolization of glucose by cells. Because of this, cells are in a state similar to nutrient deficiency. Therefore, these results may confirm our findings that nutrient deficiency reduces the *ELOVL4* expression. As for the effect of nutrient deficiency on the expression of *ELOVL1* and *ELOVL7*, the regulation of the expression of these two elongases has been very poorly understood. It is known that *ELOVL1* and *ELOVL7* expressions are upregulated by the mammalian/mechanistic target of rapamycin (mTOR) [52,58]. Under glucose deficiency, there is a decrease in the activity of this kinase [59], which may reduce the expression of *ELOVL1* and *ELOVL7*, and thus, explain our results.

As in the case of hypoxia, nutrient deficiency causes a decrease in the synthesis of very-long-chain SFA but does not affect the synthesis of MUFA and PUFA.

### 4.4. Clinical Significance of Elongases in GBM

Some elongases play a significant role in cancer processes, as indicated by the lower survival rates of GBM patients with a high expression of these enzymes. Despite this, they are very rarely studied, resulting in a low number of articles on the effect of elongase expressions on the survival of GBM patients available in the PubMed browser (https://pubmed.ncbi.nlm.nih.gov, accessed on 3 September 2022). For this reason, in this section, we used Gene Expression Profiling Interactive Analysis (GEPIA) (http://gepia.cancer-pku.cn accessed on 3 September 2022) [60] and the Pathology Atlas, which is part of the Human Protein Atlas (https://www.proteinatlas.org/humanproteome/pathology accessed on 3 September 2022) [61,62]. Both are based on data from The Cancer Genome Atlas (TCGA). GEPIA provides a range of useful information regarding the selected gene, including the effect of expression on the survival of GBM patients. The Human Protein Atlas only gives a prognosis for patients with glioma, and for this reason, the data from the Pathology Atlas may differ from GEPIA—GBM accounts only for 3/5 of all glioma cases [63].

SFA synthesis is an important fatty acid synthesis pathway in GBM, as demonstrated by the low survival rate of GBM patients when the enzymes responsible for SFA synthesis are expressed in greater levels in the tumor. In particular, according to the GEPIA portal, a worse GBM prognosis is associated with a higher expression of *ELOVL1* and *ELOVL3* [60]. Additionally, a higher expression of *ELOVL3* in glioma is associated with a worse prognosis [64], and for *ELOVL1*, it is associated with a tendency toward a worse prognosis (*p* = 0.066) [65]. On the other hand, the expression level of *ELOVL7*, the third enzyme responsible for long-chain SFA elongation, does not affect the prognosis for patients with GBM [60] or glioma [66]. The significance of *ELOVL1* and *ELOVL3* in GBM tumorigenesis remains unclear. The elevated expression of these genes is also associated with a worse prognosis for hepatocellular carcinoma [67]. At least for *ELOVL1*, a higher expression in hepatocellular carcinoma results in cancer immune evasion, including an increased expression of programmed death-ligand 1 (PD-L1) and cytotoxic T-lymphocyte-associated protein 4 (CTLA4) [67].

An important factor for cancer processes in GBM is the biosynthesis of PUFA, in particular, arachidonic acid C20:4n-6. This PUFA is processed into prostaglandin E_2_ (PGE_2_) by cyclooxygenases [68]. As PGE_2_ is responsible for radioresistance [69] and temozolomide (TMZ) resistance [70], the enzymes involved in the synthesis of arachidonic acid C20:4n-6—in particular, Elovl5 and Fads2/D6D—are also responsible for the resistance to GBM anti-tumor therapy [69]. However, GEPIA [60] and Pathology Atlas [71] do not show that a higher *ELOVL5* expression is associated with a worse or better prognosis in patients with GBM or glioma. PUFA can be further elongated to very-long-chain PUFA by *ELOVL2*. These fatty acids accumulate in the phospholipids of the cell membrane, and are followed by enhanced signal transduction from membrane receptors such as EGFR [72]. For this reason, an increase in the *ELOVL2* expression leads to a resistance to anticancer drugs that inhibit receptors with tyrosine kinase activity or drugs that reduce EGFR activity, such as lapatinib. The high importance of *ELOVL2* in tumorigenesis and resistance to treatment may confirm the association of the expression of this elongase with the prognosis for GBM patients. A higher *ELOVL2* expression in the tumor is associated with a worse prognosis for patients with GBM [29] and glioma [73]. At the same time, according to the GEPIA portal, there is no association between Elovl2 expression and prognosis for GBM patients [60].

According to the GEPIA portal, there is no correlation between *ELOVL6* expression and the prognosis for GBM patients [60]. On the other hand, Pathology Atlas showed that a higher *ELOVL6* expression in glioma tumors is associated with a worse prognosis [74]. Elovl6 is involved in the elongation of palmitoyl-CoA to stearoyl-CoA and, in this way, it participates in the synthesis of all MUFA and SFA—the structural elements of cells. The importance of this enzyme in cancer processes has been well established in hepatocellular carcinoma [75]. Here, Elovl6 is essential in cancer cell proliferation; a higher *ELOVL6* expression accompanies greater tumor growth, and thus, a greater aggressiveness of this cancer. According to the available studies, the expression level of *ELOVL4* does not affect the prognosis of patients with GBM [60] or glioma [76]. Thus, it is possible that is does not play a significant role in tumorigenesis in GBM. 

### 4.5. Limitations of the Study

In this study, the influence of hypoxia and nutrient-deficient conditions on the expression of elongases was investigated. The experiments were performed on U-87 MG cells, that is, on one cell line only. This line is widely used and is the standard model for in vitro studies of tumor mechanisms in GBM. Nevertheless, a neoplastic tumor, not only GBM, is characterized by the heterogeneity of neoplastic cells. There are also differences in tumor mechanisms between patients. For this reason, the obtained data based on the results of only one cell line may be inaccurate. However, the results presented in this article have great scientific value as a preliminary study of the role of elongases in neoplastic processes in GBM tumor. In this study, the expression of elongases at the mRNA level was analyzed. The mRNA level is closely related to the expression at the protein level. However, in some cases, the expression of the genes in question at the mRNA and protein levels may differ from one another. For this reason, the results of our study do not guarantee the observed differences in the expression of a given elongase at the protein level.

## 5. Conclusions

The results of the present study showed that:The *ELOVL1* and *ELOVL7* expressions were reduced in GBM tumors. Very-long-chain SFA synthesis was also reduced in GBM;*ELOVL2*, *ELOVL5*, *ELOVL6*, and *ELOVL7* expressions were decreased in GBM tumors in women but not in men. The expression of *ELOVL1* was upregulated in GBM tumors in women and downregulated in men. The *ELOVL4* expression was upregulated in GBM tumors in women. GBM tumors in men showed a higher expression of *ELOVL2*, *ELOVL5*, and *ELOVL6*;The demonstrated gender differences in the expression of the studied elongases may allow for the development of personalized therapy targeting fatty acid biosynthesis;Hypoxia increased the expression of *ELOVL5* and *ELOVL6* but decreased the expression of *ELOVL1*, *ELOVL3*, *ELOVL4*, and *ELOVL7* in GBM cells. Hypoxia decreased the synthesis of very-long-chain SFA but increased PUFA synthesis;Nutrient deficiency reduced the expression of *ELOVL1*, *ELOVL4*, *ELOVL5*, and *ELOVL7*. Nutrient deficiency decreased the synthesis of very-long-chain SFA but did not affect PUFA synthesis.

## Figures and Tables

**Figure 1 brainsci-12-01356-f001:**
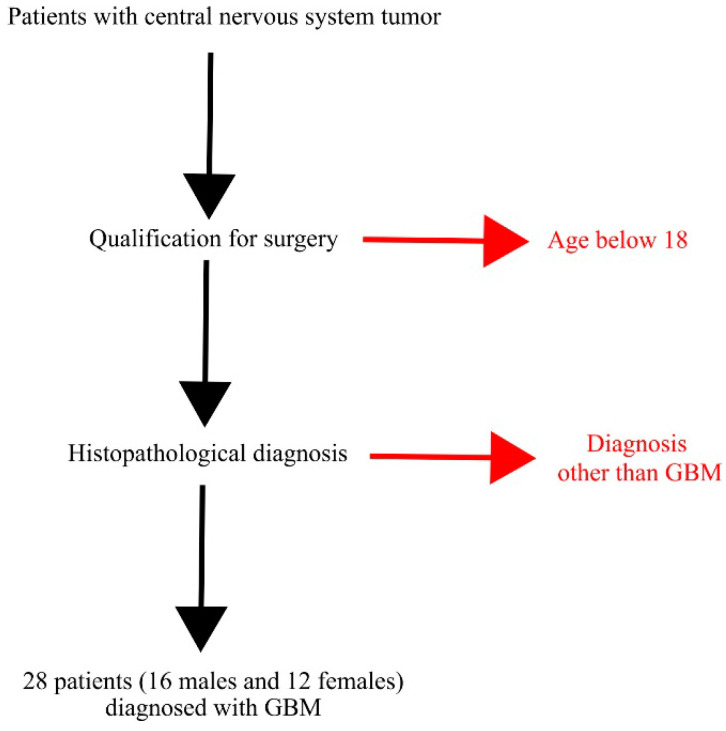
Inclusion and exclusion criteria for selecting and choosing patients from whom GBM tumor samples were analyzed.

**Figure 2 brainsci-12-01356-f002:**
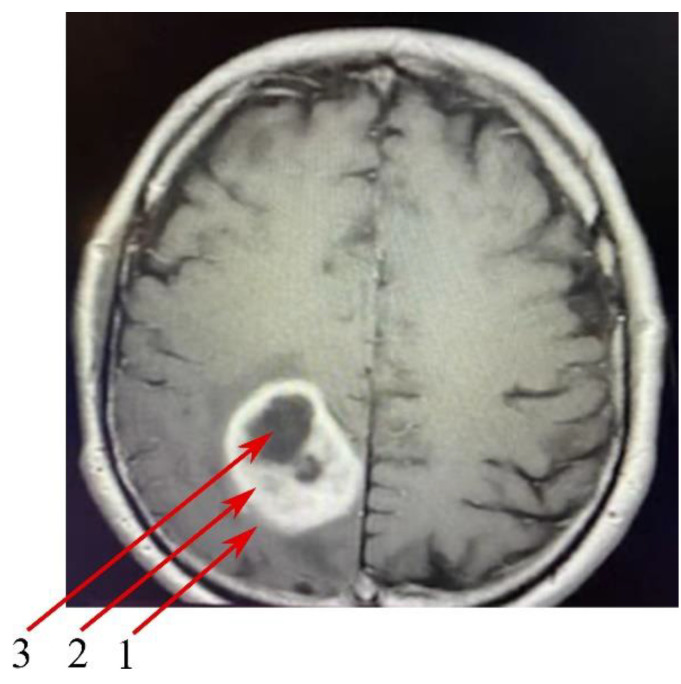
Magnetic resonance imaging (MRI) scan of the brain with the glioblastoma multiforme (GBM) tumor. The image highlights the three study zones: (1) peritumoral area, (2) enhancing tumor region and (3) tumor core. Axial, T1-weighted MRI scans of a 64-year-old patient diagnosed with GBM in the right parietal lobe. The patient presented visual impairment in the left eye and visual-spatial coordination disorders, as well as a sensory impairment on the right side.

**Figure 3 brainsci-12-01356-f003:**
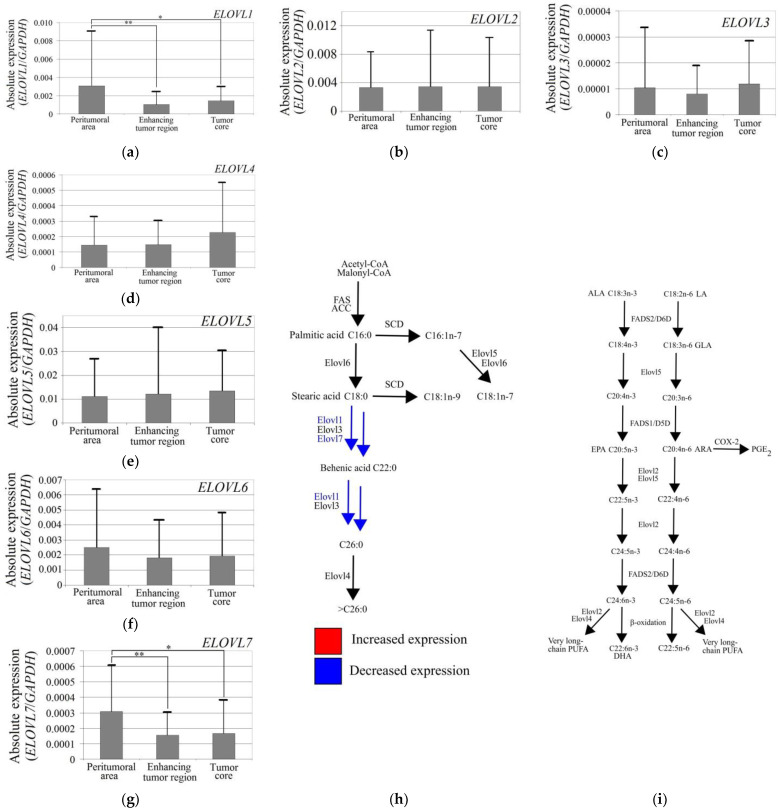
Expression levels of elongases in a GBM tumor. Expression levels of (**a**) *ELOVL1*, (**b**) *ELOVL2*, (**c**) *ELOVL3*, (**d**) *ELOVL4*, (**e**) *ELOVL5*, (**f**) *ELOVL6*, and (**g**) *ELOVL7* in the peritumoral area, enhancing tumor area and tumor core. Changes in elongase expression affect the synthesis of (**h**) MUFA, SFA, and (**i**) PUFA. *—statistically significant difference of expression of a given elongase between different regions of a GBM tumor estimated by Wilcoxon signed-rank test (*p* < 0.05). **—*p* < 0.01.

**Figure 4 brainsci-12-01356-f004:**
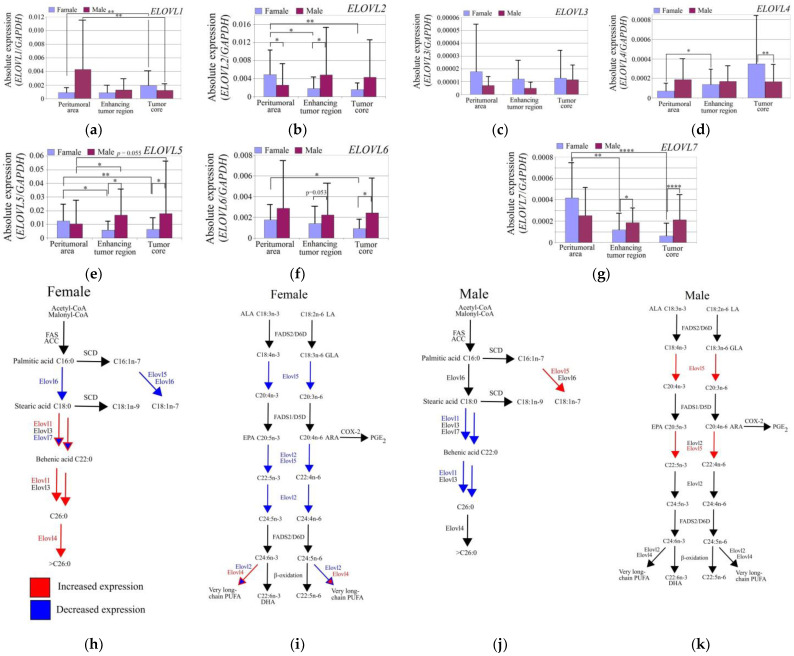
Expression levels of elongases in a GBM tumor depending on gender. Expression levels of (**a**) *ELOVL1*, (**b**) *ELOVL2*, (**c**) *ELOVL3*, (**d**) *ELOVL4,* (**e**) *ELOVL5*, (**f**) *ELOVL6*, and (**g**) *ELOVL7* in the peritumoral area, enhancing tumor area and tumor core. Changes in elongase expressions affect the synthesis of (**h**,**j**) MUFA, SFA, and (**i**,**k**) PUFA in women (**h**,**i**) and men (**j**,**k**). *—statistically significant difference of expression of a given elongase between different regions of a GBM tumor estimated by Wilcoxon signed-rank test (*p* < 0.05). **—*p* < 0.01. ****—*p* < 0.0001.

**Figure 5 brainsci-12-01356-f005:**
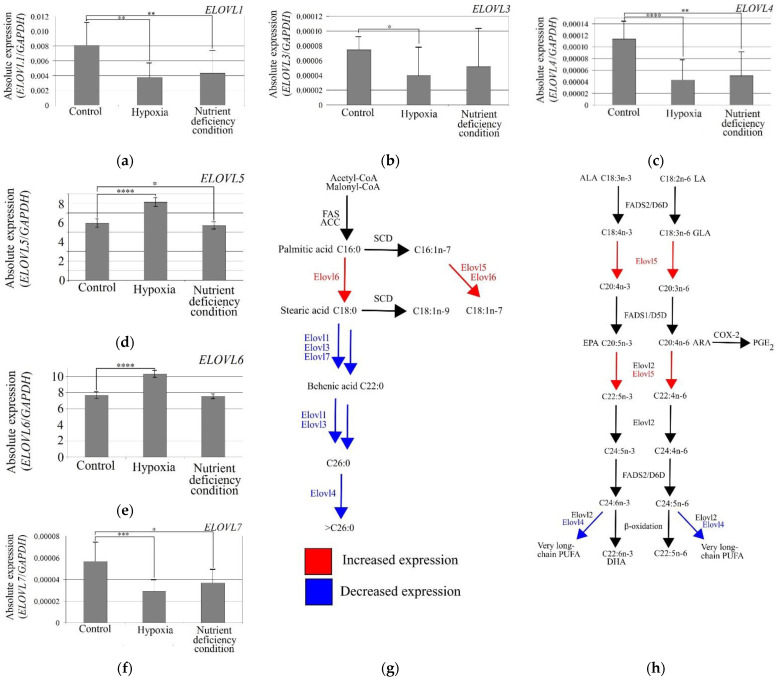
Effect of hypoxia and nutrient-deficient conditions on the expression of elongases. Effect of hypoxia and nutrient deficiency conditions on the expression of (**a**) *ELOVL1*, (**b**) *ELOVL3*, (**c**) *ELOVL4*, (**d**) *ELOVL5*, (**e**) *ELOVL6*, (**f**) and *ELOVL7* in U-87 MG cells. Hypoxia affects (**g**) the synthesis of MUFA and SFA and (**h**) PUFA. *—Statistically significant difference in the expression of a given elongase between controls estimated by Wilcoxon signed-rank test (*p* < 0.05). **—*p* < 0.01. ***—*p* < 0.001. ****—*p* < 0.0001.

**Table 1 brainsci-12-01356-t001:** The statistical characteristics of the patients.

	N	Mean	Standard Deviation	Minimum	Maximum
Age at surgery	24	60.7	12.5	36	81
Weight	24	84	19	55	130
Height	23	172	12	147	196
BMI	23	28.7	4.8	21.5	38.9
Pack-years (number of smoking years × number of packs (20 pcs.)/d	20	13.42	13.98	0	42
Men					
Age at surgery	14	60.6	11.9	41	81
Weight	14	93.8	15.4	73	130
Height	14	178	8.6	163	196
BMI	14	29.5	4.2	24.7	38.9
Pack-years (number of smoking years × number of packs (20 pcs.)/d	12	18.63	14.48	0	42
Women					
Age at surgery	12	60.8	14.1	36	79
Weight	12	70.3	15	55	95
Height	11	162	8	147	173
BMI	11	27.4	5.7	21.5	36.2
Pack-years (number of smoking years × number of packs (20 pcs.)/d	8	5.61	9.23	0	21
Hormone replacement therapy	7	5 patients were not receiving hormone therapy, 2 patients were receiving			

**Table 2 brainsci-12-01356-t002:** Correlation of the expression of the studied elongases between each other and the studied GBM tumor regions.

Studied Gene/ Location		*ELOVL1*			*ELOVL2*			*ELOVL3*			*ELOVL4*			ELOVL5			*ELOVL6*			*ELOVL7*		
		ETR	TC	PA	ETR	TC	PA	ETR	TC	PA	ETR	TC	PA	ETR	TC	PA	ETR	TC	PA	ETR	TC	PA
** *ELOVL1* **	**ETR**	1.00																				
	**TC**	0.36	1.00																			
	**PA**	−0.32	0.07	1.00																		
** *ELOVL2* **	**ETR**	0.35	−0.27	0.34	1.00																	
	**TC**	**0.44 ***	0.17	0.04	0.02	1.00																
	**PA**	0.35	−0.06	0.25	0.34	0.07	1.00															
** *ELOVL3* **	**ETR**	−0.02	−0.15	−0.02	0.37	0.29	0.13	1.00														
	**TC**	−0.44	−0.18	−0.05	**−0.49 ***	0.16	−0.33	−0.03	1.00													
	**PA**	−0.34	−0.23	−0.11	0.19	0.31	0.02	0.08	0.10	1.00												
** *ELOVL4* **	**ETR**	0.05	−0.02	0.06	0.26	0.15	−0.10	0.07	−0.10	−0.33	1.00											
	**TC**	−0.09	−0.14	0.04	−0.04	**0.37 ***	0.16	0.23	−0.02	−0.12	−0.14	1.00										
	**PA**	−0.06	0.35	−0.07	−0.09	**0.46 ***	**0.49 ***	0.02	0.36	0.03	0.36	**0.51 ***	1.00									
** *ELOVL5* **	**ETR**	0.30	0.04	**0.19**	**0.50 ***	−0.14	0.25	0.00	**0.53 ***	0.01	**0.46 ***	−0.09	0.18	1.00								
	**TC**	0.26	**0.58 ***	−0.03	−0.12	**0.65 ***	−0.22	−0.07	0.22	0.16	0.03	**0.77 ***	0.35	0.08	1.00							
	**PA**	0.13	0.16	0.24	0.16	0.01	**0.74 ***	−0.05	−0.03	0.02	−0.15	0.24	**0.51 ***	0.10	0.17	1.00						
** *ELOVL6* **	**ETR**	0.36	0.05	0.03	0.34	−0.22	0.25	0.22	0.32	−0.01	**0.53 ***	0.15	0.22	**0.72 ***	**0.73 ***	−0.13	1.00					
	**TC**	0.10	0.28	−0.12	−0.13	**0.66 ***	−0.04	0.08	0.23	0.13	0.03	**0.64 ***	0.28	0.06	**0.83 ***	0.09	0.34	1.00				
	**PA**	−0.04	0.03	0.21	−0.01	0.20	**0.61 ***	−0.19	−0.17	0.06	0.33	0.20	**0.73 ***	0.07	0.26	**0.51 ***	0.07	0.28	1.00			
** *ELOVL7* **	**ETR**	−0.12	0.17	0.00	0.22	−0.03	−0.02	0.26	0.25	−0.26	**0.60 ***	0.05	**0.40 ***	**0.54 ***	−0.03	−0.08	**0.51 ***	−0.32	0.31	1.00		
	**TC**	−0.04	−0.04	−0.13	0.01	0.31	−0.09	−0.12	−0.11	0.03	0.31	**0.60 ***	−0.23	0.29	**0.70 ***	−0.06	0.34	**0.49 ***	0.06	0.29	1.00	
	**PA**	−0.08	−0.13	0.24	−0.01	0.33	**0.58 ***	0.03	−0.30	−0.15	**−0.52 ***	0.08	0.11	0.02	0.11	**0.46 ***	−0.01	0.21	**0.58 ***	**−0.46 ***	−0.13	1.00

The values of Spearman’s rank correlation coefficients are given. * Statistically significant correlation of the expression of two genes/locations (*p* < 0.05). PA—peritumoral area; ETR—enhancing tumor region; TC—tumor core.

**Table 3 brainsci-12-01356-t003:** Mutual correlations between the expression of the studied elongases, and correlations between the expression of the elongases and GBM tumor regions in men.

Studied Gene/ Location		*ELOVL1*			*ELOVL2*			*ELOVL3*			*ELOVL4*			*ELOVL5*			*ELOVL6*			*ELOVL7*		
		ETR	TC	PA	ETR	TC	PA	ETR	TC	PA	ETR	TC	PA	ETR	TC	PA	ETR	TC	PA	ETR	TC	PA
** *ELOVL1* **	**ETR**	1.00																				
	**TC**	0.18	1.00																			
	**PA**	0.15	0.06	1.00																		
** *ELOVL2* **	**ETR**	0.32	−0.51	0.13	1.00																	
	**TC**	0.49	0.29	0.04	**0.39 ***	1.00																
	**PA**	0.41	0.11	0.36	**0.70 ***	0.46	1.00															
** *ELOVL3* **	**ETR**	−0.12	−0.03	0.06	**0.52 ***	0.28	0.10	1.00														
	**TC**	−0.39	−0.16	0.05	**−0.59 ***	0.14	**−0.64 ***	−0.35	1.00													
	**PA**	0.18	0.31	0.05	0.43	0.37	0.29	**0.60 ***	−0.21	1.00												
** *ELOVL4* **	**ETR**	−0.04	0.02	−0.03	−0.24	0.08	−0.03	0.15	−0.09	0.16	1.00											
	**TC**	0.00	**0.54 ***	−0.26	0.09	0.27	0.14	0.29	−0.04	0.19	−0.08	1.00										
	**PA**	−0.08	0.44	0.21	−0.06	**0.48 ***	**0.61 ***	0.24	−0.37	0.32	**0.55 ***	0.40	1.00									
** *ELOVL5* **	**ETR**	0.39	−0.24	0.25	0.40	0.36	**0.59 ***	0.02	−0.17	0.27	0.26	−0.27	0.29	1.00								
	**TC**	0.31	**0.67 ***	−0.02	−0.11	**0.67 ***	0.21	0.05	0.28	0.23	−0.02	**0.76 ***	0.22	0.19	1.00							
	**PA**	0.19	0.25	0.36	0.40	0.35	**0.68 ***	−0.30	−0.38	0.06	0.22	0.11	**0.57 ***	0.47	0.20	1.00						
** *ELOVL6* **	**ETR**	0.32	0.04	−0.31	0.29	**0.42 ***	**0.41 ***	−0.08	−0.32	0.44	**0.60 ***	0.14	0.46	**0.66 ***	0.32	0.26	1.00					
	**TC**	0.15	**0.49 ***	0.14	−0.09	**0.65 ***	0.08	0.15	0.14	0.19	0.15	**0.60 ***	0.27	0.03	**0.82 ***	0.16	0.31	1.00				
	**PA**	0.26	0.19	0.35	0.06	**0.54 ***	**0.72 ***	−0.11	−0.40	0.17	−0.23	0.31	**0.83 ***	0.20	0.26	**0.67 ***	0.09	0.22	1.00			
** *ELOVL7* **	**ETR**	−0.50	0.15	−0.24	−0.49	−0.33	−0.18	0.09	−0.34	0.17	**0.69 ***	0.16	**0.64 ***	0.19	−0.25	−0.30	**0.54 ***	−0.19	−0.09	1.00		
	**TC**	0.20	**0.69 ***	−0.23	−0.12	0.11	−0.02	0.04	−0.05	−0.07	−0.15	**0.76 ***	0.25	−0.28	**0.65 ***	−0.01	0.07	0.43	0.08	−0.07	1.00	
	**PA**	0.19	0.47	**0.56 ***	0.36	0.36	**0.66 ***	0.30	**−0.69 ***	0.28	0.08	0.03	**0.50 ***	0.45	0.09	**0.53 ***	0.32	0.17	**0.58 ***	0.25	0.12	1.00

The values of Spearman’s rank correlation coefficients are given. * Statistically significant correlation of the expression of two genes/locations (*p* < 0.05). PA—peritumoral area; ETR—enhancing tumor region; TC—tumor core.

**Table 4 brainsci-12-01356-t004:** Mutual correlations between the expression of the studied elongases, and correlations between the expression of the elongases and GBM tumor regions in women.

Studied Gene/ Location		*ELOVL1*			*ELOVL2*			*ELOVL3*			*ELOVL4*			*ELOVL5*			*ELOVL6*			*ELOVL7*		
		ETR	TC	PA	ETR	TC	PA	ETR	TC	PA	ETR	TC	PA	ETR	TC	PA	ETR	TC	PA	ETR	TC	PA
** *ELOVL1* **	**ETR**	1.00																				
	**TC**	0.52	1.00																			
	**PA**	0.36	0.22	1.00																		
** *ELOVL2* **	**ETR**	0.41	−0.07	0.53	1.00																	
	**TC**	0.47	0.05	0.02	0.04	1.00																
	**PA**	0.20	−0.31	0.28	0.10	0.10	1.00															
** *ELOVL3* **	**ETR**	−0.15	−0.23	0.28	0.17	0.39	0.22	1.00														
	**TC**	−0.23	−0.17	−0.20	−0.41	**0.67 ***	−0.21	0.25	1.00													
	**PA**	−0.23	−0.28	−0.35	−0.05	0.29	**−0.55 ***	0.05	**0.61 ***	1.00												
** *ELOVL4* **	**ETR**	0.24	−0.04	0.30	**0.75 ***	0.06	0.02	0.44	−0.22	−0.05	1.00											
	**TC**	−0.25	0.48	−0.15	−0.25	0.13	0.03	0.03	0.19	0.16	−0.02	1.00										
	**PA**	−0.15	**0.70 ***	−0.26	−0.31	0.53	−0.09	−0.41	0.47	0.33	−0.14	**0.75 ***	1.00									
** *ELOVL5* **	**ETR**	0.22	0.24	0.15	**0.65 ***	−0.22	0.17	0.13	**−0.74 ***	−0.35	**0.53 ***	0.09	−0.16	1.00								
	**TC**	0.25	0.55	−0.18	−0.26	**0.63 ***	−0.31	0.05	0.31	−0.04	−0.27	**0.93 ***	**0.98 ***	0.10	1.00							
	**PA**	0.20	0.24	0.09	−0.33	**0.50 ***	0.40	0.24	0.33	−0.16	−0.53	0.30	0.47	−0.13	**0.50 ***	1.00						
** *ELOVL6* **	**ETR**	0.20	0.07	0.28	**0.51 ***	−0.14	0.32	0.27	**−0.60 ***	−0.37	0.46	0.38	−0.18	**0.79 ***	0.16	−0.27	1.00					
	**TC**	0.03	0.00	−0.57	−0.27	**0.67 ***	0.37	0.24	**0.57 ***	0.11	−0.41	0.55	0.58	−0.23	**0.73 ***	0.33	−0.07	1.00				
	**PA**	0.17	−0.26	−0.12	−0.29	**0.49 ***	**0.53 ***	−0.29	0.24	−0.30	0.00	0.10	0.15	−0.26	0.15	0.07	−0.06	−0.10	1.00			
** *ELOVL7* **	**ETR**	0.09	−0.31	0.17	**0.75 ***	0.37	0.49	0.21	−0.20	−0.09	**0.64 ***	−0.23	−0.09	**0.60 ***	−0.03	0.05	0.41	−0.08	0.38	1.00		
	**TC**	0.05	−0.03	0.39	0.50	0.27	0.26	−0.09	−0.10	−0.05	0.07	−0.24	−0.05	0.54	0.17	0.43	0.36	−0.13	0.08	**0.66 ***	1.00	
	**PA**	0.38	0.36	−0.05	−0.56	0.21	0.35	−0.41	0.13	−0.40	**−0.79 ***	0.07	0.40	−0.38	0.43	**0.58 ***	−0.25	−0.07	**0.59 ***	−0.36	−0.02	1.00

The values of Spearman’s rank correlation coefficients are given. * Statistically significant correlation of the expression of two genes/locations (*p* < 0.05). PA—peritumoral area; ETR—enhancing tumor region; TC—tumor core.

**Table 5 brainsci-12-01356-t005:** Correlation between the expression of the elongases studied and the characteristics of patients.

	Parameter	Age	Body Weight	Height	BMI	Pack-Years	Age	Body Weight	Height	BMI	Pack-Years	Age	Body Weight	Height	BMI	Pack-Years
	Gender	All patients					Men					Women				
*ELOVL1*	PA	−0.04	−0.33	0.07	0.34	0.39	−0.02	−0.45	0.05	−0.31	0.18	−0.13	**−0.67 ***	−0.40	−0.55	0.63
	ETR	−0.04	0.16	0.21	−0.01	0.27	0.17	0.30	−0.06	0.35	0.29	−0.53	−0.56	0.14	−0.57	0.48
	TC	−0.19	−0.10	0.06	−0.17	−0.26	0.13	0.14	0.00	0.26	**−0.78 ***	**−0.63 ***	**−0.75 ***	0.04	**−0.68 ***	**0.81 ***
*ELOVL2*	PA	−0.01	−0.23	**−0.47 ***	0.26	−0.16	0.02	0.20	−0.29	**0.51 ***	−0.07	−0.19	−0.06	−0.11	0.29	0.26
	ETR	−0.24	0.02	0.17	0.05	**0.45 ***	0.08	−0.24	−0.40	0.20	**0.77 ***	**−0.46 ***	−0.16	0.46	−0.34	−0.17
	TC	0.19	0.07	−0.11	0.01	−0.21	0.13	0.13	−0.24	**0.41 ***	0.00	0.17	**−0.48 ***	−0.11	−0.32	−0.35
*ELOVL3*	PA	0.02	−0.16	−0.25	−0.15	−0.28	−0.06	−0.22	−0.27	0.02	0.35	0.14	0.19	0.11	−0.05	−0.54
	ETR	−0.01	−0.36	−0.31	−0.34	−0.29	0.20	−0.54	−0.46	−0.15	0.26	0.07	−0.08	0.16	−0.36	0.00
	TC	0.21	**−0.44 ***	−0.35	**−0.46 ***	−0.23	0.030	−0.47	−0.14	**−0.57 ***	0.01	0.44	−0.38	−0.41	−0.22	−0.34
*ELOVL4*	PA	−0.38	−0.09	0.03	−0.24	**−0.52 ***	−0.52	0.27	0.38	0.20	−0.42	−0.01	−0.25	−0.14	−0.29	0.00
	ETR	−0.38	0.24	**0.63 ***	−0.26	0.01	−0.05	0.26	0.55	−0.17	−0.12	−0.49	−0.29	0.24	−0.47	−0.15
	TC	−0.12	−0.41	−0.42	−0.32	−0.40	−0.04	−0.30	−0.41	0.01	**−0.69 ***	−0.04	−0.49	−0.03	−0.66	0.34
*ELOVL5*	PA	−0.16	−0.23	−0.32	0.05	−0.06	0.08	0.27	0.11	0.35	0.23	0.15	−0.12	−0.41	0.05	0.37
	ETR	−0.36	**0.43 ***	0.09	**0.58 ***	0.27	0.14	0.23	−0.13	**0.39 ***	0.23	**−0.61 ***	**0.82 ***	0.01	**−0.45 ***	0.15
	TC	−0.05	0.17	0.14	−0.19	0.00	−0.08	0.10	−0.19	0.30	−0.33	0.05	**−0.42 ***	0.25	**−0.71 ***	0.06
*ELOVL6*	PA	−0.21	0.05	0.13	0.18	−0.08	−0.35	0.13	0.29	0.10	−0.19	−0.26	−0.26	−0.60	0.38	−0.11
	ETR	−0.15	0.22	0.36	−0.02	0.24	0.32	0.24	−0.22	**0.38 ***	−0.33	−0.34	−0.17	−0.07	**−0.52 ***	0.22
	TC	0.25	0.02	−0.15	−0.12	−0.11	0.29	−0.20	−0.30	−0.01	−0.09	0.29	−0.05	0.00	−0.21	−0.23
ELOVL7	PA	0.12	−0.04	−0.35	0.22	−0.28	0.10	**0.55 ***	−0.07	**0.76 ***	−0.19	0.13	−0.13	−0.46	0.33	0.11
	ETR	**−0.57 ***	0.27	0.46	0.18	−0.28	−0.50	0.50	0.36	0.25	−0.71	−0.55	0.01	0.36	−0.02	**−0.73 ***
	TC	−0.10	0.40	0.45	0.05	−0.15	−0.04	0.12	0.02	0.12	**−0.78 ***	−0.37	−0.15	0.33	−0.09	−0.39

The values of Spearman’s rank correlation coefficients are given. * Statistically significant correlation of the expression of two genes/locations (*p* < 0.05). PA—peritumoral area; ETR—enhancing tumor region; TC—tumor core.

## Data Availability

The data presented in this study are available on request from the corresponding author.

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
