# Peer review of "Glioblastoma Multiforme Tumors in Women Have a Lower Expression of Fatty Acid Elongases ELOVL2, ELOVL5, ELOVL6, and ELOVL7 than in Men"

_brainsci, 2022, doi:10.3390/brainsci12101356_

Round 1

Reviewer 1 Report (Previous Reviewer 2)

The authors have satisfactorily revised that the subject matter of this work is acceptable.

Author Response

We greatly appreciate thorough and kind review. 

Reviewer 2 Report (Previous Reviewer 1)

The authors presented the manuscript "Glioblastoma multiforme tumors in women have lower expression of fatty acids elongases ELOVL2, ELOVL5, ELOVL6, and ELOVL7 than in men”

The topic is of general interest to the scientific community, yet the reviewer has several concerns regarding the manuscript.

No changes were made to the manuscript based on this reviewer’s previous comments. 

Comment one, was that the sample size was to low, and the inclusion of a power analysis should have been performed. 

Comment two, was that validation experiments should be performed in a second cell line

Comment three,  relative expression calculations in qPCR are not appropriately addressed. They look more as normalized (not relative).

Comment four, not sure regarding the statistics, yet in figure 4 many standard deviations are 5 to six times greater than mean value

Comment five, is there any data that supports a claim for elongases supporting 5-yer survival, drug response, relapse rate, etc… 

Comment six, what is the protein expression of elognase and isoforms in cancer cell lines 

Author Response

We are very grateful to the Reviewer for a thorough and kind review.

Comment one, was that the sample size was to low, and the inclusion of a power analysis should have been performed. 

Power analysis was calculated and corrected and Statistical Methods was supplemented. 

The study with 28 patients (16 men and 12 women) had sufficient statistical power to detect with 80% probability the following real effect sizes for the analyzed quantitative variables: difference between tumor zones or culture conditions (paired measurements) corresponding to 0.55 standard deviations of the difference between paired measurements for the analyzed parameter; difference between males and females (independent groups) corresponding to 1.2 standard deviations of the analyzed parameter; correlation between analyzed parameters corresponding to the value of correlation coefficient equal to ±0.50.

Comment two, was that validation experiments should be performed in a second cell line

We thank the reviewer for the valuable comment. In our study, we showed that hypoxia alters the expression of elongases in U-87 MG cells. This line is widely used as a standard model for in vitro studies of tumorigenic mechanisms in GBM. At the same time, we understand the reviewer's remark, and for this reason we have included a note about performing in vitro experiments on only one line, which may not be sufficient to fully understand the role of elongases in the tumor processes studied. We also would like to point out that we currently do not have the funds that would allow us to extend the research to a new line and conduct new experiments. After obtaining the funds, it will certainly be the direction of our further research, in line with the reviewer's remark.

4.5 Limitations of the study

In this study, the influence of hypoxia and nutrient-deficient conditions on the expression of elongases was investigated. The experiments were performed on U-87 MG cells, that is, on one cell line only. This line is widely used and is the standard model for in vitro studies of tumor mechanisms in GBM. Nevertheless, a neoplastic tumor, not only GBM, is characterized by the heterogeneity of neoplastic cells. There are also differences in tumor mechanisms between patients. For this reason, the obtained data based on the results of only one cell line may be inaccurate. However, the results presented in this article have great scientific value as a preliminary study of the role of elongases in neoplastic processes in GBM tumor. In this study, the expression of elongases at the mRNA level was analyzed. The mRNA level is closely related to the expression at the protein level. However, in some cases, the expression of the genes in question at the mRNA and protein levels may differ from one another. For this reason, the results of our study do not guarantee the observed differences in the expression of a given elongase at the protein level.

Comment three,  relative expression calculations in qPCR are not appropriately addressed. They look more as normalized (not relative).

We would like to thank reviewer #2 for this comment. Indeed, we have now presented in Results and described in Materials and Methods qPCR results in form of the normalized or absolute expression. This was calculated as 2ˆdCt, where dCt = gene Ct - GAPDH Ct. Accordingly, we have changed the “y” axis name to “Absolute expression” in Figures 3, 4 and 5. Additionally we have deleted “relative” and improved the description of the method.

Comment four, not sure regarding the statistics, yet in figure 4 many standard deviations are 5 to six times greater than mean value

We agree with the reviewer that a large standard deviation looks bad in a scientific article. However, there are small standard deviations in in vitro experiments. Cells from different replicates of the experiment are the same cell line that was cultured under the same conditions. A large standard deviation is found in our study of GBM patients. Each patient is genetically different, as well as different in terms of lifestyle, work, diet, number of cigarettes smoked, etc. Also given genetic mutations in GBM tumors are with a certain probability. All these factors cause a scattering of results in the analysis of patient samples which sometimes leads to large standard deviations. This, however, can also be an important result of our article.

Comment five, is there any data that supports a claim for elongases supporting 5-yer survival, drug response, relapse rate, etc… 

A description of the clinical significance of elongases has been added.

4.4. Clinical significance of elongases in GBM

Some elongases play a significant role in cancer processes, as indicated by the lower survival rates of GBM patients with a high expression of these enzymes. Despite this, they are very rarely studied, resulting in a low number of articles on the effect of elongase expression on the survival of GBM patients available in the PubMed browser (https://pubmed.ncbi.nlm.nih.gov accessed 03 Sep 2022). For this reason, in this section we used Gene Expression Profiling Interactive Analysis (GEPIA) (http://gepia.cancer-pku.cn accessed 03 Sep 2022) [60] and the Pathology Atlas, part of the Human Protein Atlas (https://www.proteinatlas.org/humanproteome/pathology accessed 03 Sep 2022) [61,62]. Both are based on data from The Cancer Genome Atlas (TCGA). GEPIA provides a range of useful information regarding the selected gene, including the effect of expression on the survival of GBM patients. The Human Protein Atlas only gives a prognosis for patients with glioma, and for this reason the data from the Pathology Atlas may differ from GEPIA – GBM accounts only for 3/5 of all glioma cases [63].

SFA synthesis is an important fatty acid synthesis pathway in GBM, as demonstrated by the low survival rate of GBM patients when the enzymes responsible for SFA synthesis are expressed in greater levels in the tumor. In particular, according to the GEPIA portal, a worse GBM prognosis is associated with higher expression of ELOVL1 and ELOVL3 [60]. Also, higher expression of ELOVL3 in glioma is associated with a worse prognosis [64] and for ELOVL1 it is associated with a tendency toward a worse prognosis (p=0.066) [65]. On the other hand, the expression level of ELOVL7, the third enzyme responsible for long-chain SFA elongation, does not affect the prognosis for patients with GBM [60] or glioma [66]. The significance of ELOVL1 and ELOVL3 in GBM tumorigenesis remains unclear. The elevated expression of these genes is also associated with a worse prognosis for hepatocellular carcinoma [67]. At least for ELOVL1, a higher expression in hepatocellular carcinoma results in cancer immune evasion, including increased expression of programmed death-ligand 1 (PD-L1) and cytotoxic T-lymphocyte-associated protein 4 (CTLA4) [67].

An important factor for cancer processes in GBM is the biosynthesis of PUFA, in particular arachidonic acid C20:4n-6. This PUFA is processed into prostaglandin E2 (PGE2) by cyclooxygenases [68]. As PGE2 is responsible for radioresistance [69] and temozolomide (TMZ) resistance [70], the enzymes involved in the synthesis of arachidonic acid C20:4n-6, in particular Elovl5 and Fads2/D6D, are also responsible for resistance to GBM anti-tumor therapy [69]. However, GEPIA [60] and Pathology Atlas [71] do not show that higher ELOVL5 expression is associated with worse or better prognosis in patients with GBM or glioma. PUFA can be further elongated to very-long chain PUFA by ELOVL2. These fatty acids accumulate in the phospholipids of the cell membrane, which is followed by enhanced signal transduction from membrane receptors such as EGFR [72]. For this reason, an increase in ELOVL2 expression leads to resistance to anticancer drugs that inhibit receptors with tyrosine kinase activity or drugs that reduce EGFR activity, such as lapatinib. The high importance of ELOVL2 in tumorigenesis and resistance to treatment may confirm the association of the expression of this elongase with prognosis for GBM patients. Higher ELOVL2 expression in the tumor is associated with a worse prognosis for patients with GBM [29] and glioma [73]. At the same time, according to the GEPIA portal, there is no association between Elovl2 expression and prognosis for GBM patients [60]. According to the GEPIA portal, there is no correlation between ELOVL6 expression and the prognosis for GBM patients [60]. On the other hand, Pathology Atlas showed that higher ELOVL6 expression in glioma tumors is associated with a worse prognosis [74]. Elovl6 is involved in the elongation of palmitoyl-CoA to stearoyl-CoA and in this way it participates in the synthesis of all MUFA and SFA, the structural elements of cells. The importance of this enzyme in cancer processes has been well established in hepatocellular carcinoma [75]. Here, Elovl6 is essential in cancer cell proliferation; a higher ELOVL6 expression accompanies greater tumor growth and thus a greater aggressiveness of this cancer. According to available studies, the expression level of ELOVL4 does not affect the prognosis of patients with GBM [60] or glioma [76]. Thus, it is possible that is does not play a significant role in tumorigenesis in GBM.

Comment six, what is the protein expression of elognase and isoforms in cancer cell lines 

In our study, we analyzed the expression of elongases at the mRNA level. Expression at the mRNA and protein levels are related to each other. A greater number of mRNA increases the probability of translation of a given gene. For this reason, an elevated mRNA level of a particular elongase increases the protein level of that elongase, and analysis at the mRNA level is sufficient to understand most of the molecular processes involved in the expression of the genes in question. Nevertheless, understanding the reviewer's reservations, we have now included an appropriate paragraph in the section on the limitations of the study.

We also would like to point out that we currently do not have the funds that would allow us to extend the new experiments. After obtaining the funds, it will certainly be the direction of our further research, in line with the reviewer's remark.

Round 2

Reviewer 2 Report (Previous Reviewer 1)

the reviewer thanks the authors for providing additional supportive evidence and scientific literature that supports their claims

This manuscript is a resubmission of an earlier submission. The following is a list of the peer review reports and author responses from that submission.

Round 1

Reviewer 1 Report

The authors describe the role of a family of elongases in GBM. While the authors have done a decent job of describing the expression differences of these ELOV elongases in certain tumor regions, overall the data is unconvincing, and there seems to be little to difference between males and females. 

Furthermore, besides slight differential expression patterns of ELOVs in tumor areas, it is not clear how this information relates to any clinical benefit (overall survival, drug response, activation of a particular fatty acid metabolism pathway)

Reviewer 2 Report

The study by Korbecki et al. investigated the expression of elongases of long-chain fatty acid family members ELOVL2, ELOVL5 and ELOVL6 in GBM tumor samples from 28 patients (16 men and 12 women). The results showed that saturated fatty acids (SFA) and polyunsaturated fatty acids (PUFA) in GBM tumors in men may be higher than in women. The overall data presented by this manuscript is too preliminary.

Following are some concerns which need to be addressed by the authors:

  1. Overall sample size for the current study is low, thus the results might not be reliable given inadequate power of the study. An a priori power calculation would have been useful.
  2. Entire experiment is performed with only one cell line. To generalize the finding and to eliminate the possibility that this is not one cell phenomena, authors need to use more glioblastoma multiforme cell lines in these experiments.
  3. Please dampen the conclusions, their claims are too strong for the evidences.